# Fabrication and DC-Bias Manipulation Frequency Characteristics of AlN-Based Piezoelectric Micromachined Ultrasonic Transducer

**DOI:** 10.3390/mi14010210

**Published:** 2023-01-14

**Authors:** Tao Li, Le Zhang, Wenping Geng, Jian He, Yongkang Rao, Jiabing Huo, Kunxian Yan, Xiujian Chou

**Affiliations:** Science and Technology on Electronic Test and Measurement Laboratory, North University of China, Taiyuan 030051, China

**Keywords:** piezoelectric micromachined ultrasonic transducer (PMUT), aluminum nitride (AlN), microelectromechanical system (MEMS) fabrication, DC bias, frequency characteristics

## Abstract

Due to their excellent capabilities to generate and sense ultrasound signals in an efficient and well-controlled way at the microscale, piezoelectric micromechanical ultrasonic transducers (PMUTs) are being widely used in specific systems, such as medical imaging, biometric identification, and acoustic wireless communication systems. The ongoing demand for high-performance and adjustable PMUTs has inspired the idea of manipulating PMUTs by voltage. Here, PMUTs based on AlN thin films protected by a SiO_2_ layer of 200 nm were fabricated using a standard MEMS process with a resonant frequency of 505.94 kHz, a −6 dB bandwidth (BW) of 6.59 kHz, and an electromechanical coupling coefficient of 0.97%. A modification of 4.08 kHz for the resonant frequency and a bandwidth enlargement of 60.2% could be obtained when a DC bias voltage of −30 to 30 V was applied, corresponding to a maximum resonant frequency sensitivity of 83 Hz/V, which was attributed to the stress on the surface of the piezoelectric film induced by the external DC bias. These findings provide the possibility of receiving ultrasonic signals within a wider frequency range, which will play an important role in underwater three-dimensional imaging and nondestructive testing.

## 1. Introduction

Ultrasonic imaging has dominated the field of biomedical imaging, especially Intravascular Ultrasound (IVUS), which is an outstanding technology that provides vital imaging information for the diagnosis of coronary arterial diseases [1,2]. An ultrasound imaging system can provide real-time, high-resolution images, and it is a more affordable medical imaging solution than magnetic resonance imaging and computed tomography. Ultrasonic imaging has also been widely used in other areas of research beyond medical applications, including rangefinders [3,4], fingerprint sensing [5,6], fluid density sensing [7,8], communication links [9], underwater 3D imaging [10,11], nondestructive testing [12], and wireless power supply for implantable microdevices [13]. As one of the typical ultrasonic imaging devices, piezoelectric micromechanical ultrasound transducers (PMUTs) are particularly attractive due to their affordability, compact size, and compatibility with Complementary Metal Oxide Semiconductor (CMOS) manufacturing processes [5,7].

The characteristic structure of a PMUT is constituted by a piezoelectric film and a passive layer, which is driven by an electric field for the ejector or an acoustic signal for the acceptor. With the help of the piezoelectric effect, PMUTs can achieve the interconversion of electrical and acoustic energy. When an AC signal is applied, the transverse internal stress generated by the piezoelectric layer will cause the diaphragm to vibrate and generate ultrasonic waves. In contrast, when the ultrasonic waves hit the diaphragm, an electrical signal will be detected by the electrodes [14]. Hence, piezoelectric materials play a primary role in the performance of PMUT devices. Among the most studied piezoelectric materials, such as PZT [6,15], AlN [5], AlScN [4], ZnO [16], PVDF [17], and LiNbO3 [9,18], AlN is a competitive candidate due to the excellent compatibility between CMOS and MEMS micromachining. To further improve the performance of the transmit and receive sensitivity of a PMUT, the structure of the PMUT must be optimized. Muralt et al. [15] found that the Si and SiO2 passive layers should be thicker than the piezoelectric films to obtain a high electromechanical coupling coefficient so that the neutral axis is outside the piezoelectric film. Choi et al. [19,20] found that the fundamental frequencies increased with the thickness of the piezoelectric layers, and the membrane with an aspect ratio equal to 1 had the highest effective coupling coefficient. Smyth et al. [21] optimized the top-electrode coverage with a 60–70% area ratio of a circular thin-film PMUT. However, compared to conventional piezoelectric ultrasonic transducers operating in d_33_ mode and capacitive micromachined ultrasonic transducers (CMUTs), PMUTs still have the fatal disadvantage of a smaller bandwidth (BW), resulting in a limited frequency range for work [22,23], especially in the field of high axial imaging resolution. As the key performance parameters of PMUTs, the quality factor (Q) and −6 dB BW are especially vital to the application of ultrasonic imaging. Pop et al. [9] designed PMUT arrays with different sizes of top electrodes based on X-cut LiNbO3 that included serial elements centered on different resonant frequencies. These frequencies are so closely spaced that they can cover the extensive desired bandwidth. Shelton et al. [24] proposed the use of Deep Reactive Ion Etching (DRIE) to create a tube resonator extending from the back of the wafer to the bottom of the PMUT membrane to increase the real acoustic impedance detected by the PMUT, which results in a large increase in the bandwidth. Hajati et al. [25] designed a combination of micromachined dome-shaped piezoelectric resonators, in which they obtained an adjustable frequency response and a higher BW by modifying the PMUT dimensions. Rozen et al. [26] doubled the bandwidth by designing patterned wedge-shaped ribs on the PMUT to reduce the mass of the diaphragm while increasing the stiffness. Wang et al. [27] proposed the use of a piston diaphragm on the central circular diaphragm to modify the PMUT, resulting in a remarkable improvement in transmission sound pressure and the frequency bandwidth of the modified transducer. Although these methods could extend the BW to some extent, they have the drawback of greatly increasing the complexity of the PMUT design or fabrication. More importantly, the resonant frequency cannot be manipulated once the structure of the devices is given, which will tremendously restrict the applications of PMUT.

In this study, PMUT devices based on AlN films were fabricated with BW enlargement and resonant frequency manipulation using a DC bias. It was found that the center frequency, electromechanical coupling coefficient, and −6 dB bandwidth of the device vary almost linearly when a DC bias voltage from −30 V to 30 V was applied to the PMUT. A modification of 4.08 kHz for the resonant frequency and a bandwidth enlargement of 60.2% was achieved due to the function of the DC bias. These results are promising for suppressing the center frequency shift and preparing high-bandwidth devices, which could be used in the fields of ultrasonic imaging and nondestructive testing.

## 2. Design and Fabrication

The designed PMUT device has a diameter of 400 µm with a cavity of 360 µm. The vibrating film of the PMUT consists of a 1 µm AlN thin-film piezoelectric layer, a 5 µm silicon passive layer, and a 1 µm silicon dioxide stop layer. The designed upper and lower electrodes are 500 nm Al and 200 nm Mo, respectively.

The simulation model was built using the finite-element-based multiphysics model design simulation software COMSOL Multiphysics 5.5. The basic principle of the finite element analysis method is to divide each domain into small subdomain meshes. The quality of the grid division primitive directly determines the computational accuracy of the finite element simulation. Usually, the finer the mesh is, the more accurate the calculation results will be. However, the dense mesh will make the computation very complex. Since the simplified geometric model is relatively regular, a free triangular mesh was used to construct the vibrating film. The thickness of the upper electrode and the lower electrode should be less than the thickness of the entire vibrating film, which will have little effect on the vibration performance of the device. Therefore, the electrodes were equated to a plane that is circular. Based on the basic structural design of the PMUT device, aluminum nitride, silicon, and silicon dioxide required for the simulation were selected from the material library. After completing the material setup, physical field structure distribution, and network division, the PMUT characteristic frequency was simulated. The front four vibration modes of the PMUT are ranked in Figure 1. The first-order mode is displayed in Figure 1a, in which the entire film vibrates up and down with only one peak at the center of the film. The second-order mode of the device is divided into two parts along the diameter with two peaks in opposite directions, as shown in Figure 1b. The film is divided equally into four parts along the radial direction for the third-order mode, given in Figure 1c, in which adjacent regions show opposite vibration trends. As for the fourth-order mode shown in Figure 1d, the center and surrounding parts of the film vibrate in opposite directions. Usually, only the first-order vibration mode results, and the resonant frequency of the designed PMUT device is 506.21 kHz. 

The PMUT device was fabricated on a 4-inch silicon-on-insulator (SOI) wafer, and a naked device is shown in Figure 2. The whole process is given in Figure 2b–j, from the single-sided polished SOI wafer to the AlN-based PMUT device with electrodes. Figure 2b shows an SOI wafer custom-made by Suzhou Research Materials Microtech Co., Ltd. Subsequently, 200 nm of Mo and 1 µm of AlN were sputtered on SOI at Suzhou Institute of Nano-Tech and Nano-Bionics, CAS, as shown in Figure 2c. The wet etching technique was used as the method of achieving micropatterning of the AlN thin film [28]. Since the commonly used photoresist could not be used as a stop layer in this case, the SiO_2_ layer deposited by plasma-enhanced chemical vapor deposition, as shown in Figure 2d, was used as a mask for AlN. SiO_2_ etching was processed using a Reactive Ion Etching (RIE) device.

The wafer was placed in a 20% tetramethylammonium hydroxide (TMAH) solution for 30 min to complete the AlN patterning, as shown in Figure 2e. To prevent AlN from being destroyed by the alkaline developer in later processes, SiO_2_ films were retained. The next step was to complete the electrode structure patterning. Figure 2f shows the etching of metal Mo using the plasma etching method to obtain laser-cut channels. Al was used for the top electrode of the PMUT, and the etching of the upper electrode window of SiO_2_ was also completed using RIE equipment before the top electrode was prepared. Al was formed by peeling the structure, as in Figure 2g. Figure 2h shows the metal Au pad fabrication, where a layer of Au was deposited to connect the PMUT to the external PCB; metal sputtering was performed directly on the wafer, sputtering 20 nm Cr and 100 nm Au, and the pattern was formed by plasma etching. The bottom cavity preparation process, exhibited in Figure 2i, was completed using RIE and DRIE to etch the bottom SiO_2_ and Si, respectively. Finally, the devices needed to be released from the 4-inch wafer. Since mechanical scribing is more destructive to the wafer and tends to damage the device structure, laser cutting was used to separate the device, as shown in Figure 2j. A well-fabricated PMUT cell and PMUT arrays with spaces of 500 µm are displayed in Figure 2k,l.

## 3. Results and Analysis

The mechanical vibration of the PMUT was measured by a Polytec MSA-400 Laser Doppler Vibrometer (LDV), as shown in Figure 3a. Under a driving voltage of 5 V, a resonant frequency of 505.94 kHz and a maximum displacement of the center point of 12 nm were obtained, as shown in Figure 3b. The surface profile of the PMUT is shown in the inset of Figure 3b, which is consistent with the first-order vibration mode shown in Figure 1a. To characterize the electrical performance of the PMUT cell, an impedance analyzer (KEYSIGHT E4990A) was used to test the impedance and phase spectrum of the PMUT device, as shown in Figure 3c,d, with a resonant frequency of 516.27 kHz. It can be found that the LDV result is approximately the same as the simulation result. The deviation of 12.84 kHz for the resonant frequency calculated by the LDV and impedance analyzer is due to the presence of parasitic capacitance and inductance during the test using the impedance analyzer.

The resonance frequency of the PMUT is significantly affected by the manufacturing process, such as residual stresses and etching precision. The resonant frequency $f_{r}$ of the PMUT can be expressed as [29]
(1)$$f_{r}=\frac{0.60\cdot 2t}{r^{2}}\sqrt{\frac{Y}{\rho }[1+\frac{0.8\sigma }{Y}{\left(r/t\right)}^{2}]}$$
where r is the radius of the PMUT diaphragm, Y is the average plate modulus, ρ is the average density, t is the total layer thickness, and σ is the stress induced by the DC bias voltage. As can be seen in the equation, the resonant frequency of the PMUT is sensitive to σ when the ratio (r/t) is large [30]. It provides a way to manipulate the resonant frequency via the modification of σ by applying a voltage to the piezoelectric films of PMUT devices, which is also beneficial to extend the bandwidth. 

The frequency response curves of the PMUT under a DC bias voltage range from −30 V to 30 V with a step width of 10 V are displayed in Figure 4. The overlap of the frequency response curves of the PMUT arrays can be clearly observed. If PMUTs modulated with different bias voltages are reasonably coupled together, the bandwidth of PMUTs can be increased to a great extent [9,27,31], as shown in Figure 4b. The fabricated PMUT device has a −6 dB bandwidth of 6.59 kHz. The resonant frequency of the PMUT is shifted after applying a DC bias, and there is an overlap in the bandwidth when the step width is reduced to a certain level. A bandwidth of 10.56 kHz can be gained between −30 V and 30 V for the designed PMUT, which is increased by 60.2% compared to the unbiased voltage. This significant increase in bandwidth is noteworthy and means that PMUT devices can receive a wider frequency range of ultrasonic signals, which will play an important role in underwater 3D imaging and nondestructive testing. A high-bandwidth ultrasonic transducer is beneficial to improve the consistency of the signals received by the device. In addition, high-bandwidth ultrasonic transducers receive ultrasound waves with less residual vibration, which is beneficial for improving the imaging distance resolution.

The DC bias effect on the performance of the PMUT is displayed in Figure 5. The resonant frequency varies linearly with the bias voltage in a range between −30 V and +30 V, as shown in Figure 5a, in which the tendency of the curves is the same for both the LDV and the impedance analyzer. The resonant frequency is manipulated by 4.08 kHz, corresponding to a sensitivity of 68 Hz/V according to the LDV results, which can also be explained by the equation of the resonant frequency. For the impedance analyzer results, a modification of 4.98 kHz with a sensitivity of 83 Hz/V was obtained. 

The electromechanical coupling factor $k^{2}$ is used to express the conversion rate between electrical and mechanical energy, or vice versa. PMUTs are thin-film piezoelectric devices operating in the 31 mode, and their coupling coefficient $k_{31}^{2}$ is defined as follows [21]:(2)$$k_{31}^{2}=\frac{2d_{31}^{2}\cdot Y_{p}}{\varepsilon_{33}^{T}\left(1-\nu_{p}\right)}$$
where $\varepsilon_{33}^{T}$ is the relative permittivity at constant stress, while $\nu_{p}$ and $Y_{p}$ are the Poisson’s ratio and Young’s modulus of the piezoelectric material, respectively. PMUTs are usually configured with passive layers, such as silicon or silicon dioxide. The residual stresses produced during the fabrication process of the films will lead to a very complex analysis of the general form of the coupling coefficient of PMUTs. Hence, reactance is generally used to characterize the effective coupling coefficient of PMUTs. The equation for the effective coupling coefficient $k_{\mathrm{eff}}^{2}$ using the measured reactance is given by [14]
(3)$$k_{\mathrm{eff}}^{2}=\frac{C_{m}}{C_{0}+C_{m}}=\frac{f_{a}^{2}-f_{r}^{2}}{f_{a}^{2}}$$
where $C_{m}$ is the motional capacitance, $C_{0}$ is the passive capacitance, $f_{a}$ is the antiresonant frequency, and $f_{r}$ is the resonant frequency. As can be calculated from Figure 3d, the $f_{a}$ and $f_{r}$ of the PMUT are 518.87 kHz and 516.27 kHz, respectively. According to the equation, $k_{\mathrm{eff}}^{2}$ is calculated to be 0.97% for this device. The resonance frequency of the PMUT is significantly affected by the process, such as residual stresses and etching precision. 

As shown in Figure 5b, the $k_{eff}^{2}$ of the device as a function of DC bias is almost constant and exhibits excellent stability during the process of DC voltage manipulation. The maximum displacement is associated with the DC bias due to the nature of piezoelectricity.

Q and BW are crucial parameters for measuring the dynamic performance of PMUTs. The relationship can be expressed by the following equation:(4)$$Q=\frac{\sqrt{3}f_{r}}{\mathrm{BW}}=\frac{\sqrt{3}f_{r}}{f_{2}-f_{1}}$$
where $f_{2}$ and $f_{1}$ are the frequencies above and below $f_{r}$, at which the power is a quarter of the maximum power [21,32], reflected in the frequency response at a position half of the maximum displacement. According to the LDV results of PMUT, the −6 dB BW and Q of the PMUT without the DC bias voltage can be calculated as 6.59 kHz and 130.79. As can be calculated from Figure 5c, the effects of variation in the DC bias voltage on the −6 dB bandwidth and Q value are almost linear. Compared with the non-biased condition, the −6 dB BW increased to 7.10 kHz at −30 V and declined to 6.33 kHz at 30 V, showing a 6% increase or attenuation. Since Q is inversely proportional to BW, a decrease of 7.827 and an increase of 8.244 at −30 V and 30 V were found for Q. The estimated sensitivity of −6 dB BW is –12.83 Hz/V, whereas the sensitivity of Q is 0.27/V. The reason for this variation might be the additional stress and loss due to the high voltage. Similar to the structure of the resonators, the quality factor of the PMUT is intrinsically affected by air damping, anchor loss, thermoelastic dissipation, and surface loss [33]. Hence, the stresses generated by the DC bias voltage change the film support structure, leading to changes in anchor losses. 

## 4. Conclusions

In this paper, the simulation model and MEMS manufacturing process of a PMUT based on AlN were presented in detail, and the DC bias effect on the performance of the device was systematically studied. The resonant frequency of the designed PMUT device is 506.21 kHz. The resonant frequency of the prepared PMUT device is 505.94 kHz with a $k_{eff}^{2}$ of 0.97% and a −6 dB BW of 6.59 kHz. When a DC bias voltage is applied to the PMUT, the resonant frequency and bandwidth can be efficiently manipulated. A modification of 4.08 kHz for the resonant frequency of the device and a bandwidth enlargement of 60.2% could be obtained when a DC bias voltage of −30 to 30 V was applied, corresponding to a maximum resonant frequency sensitivity of 83 Hz/V. This is vital to practical applications, especially in environments with wider-frequency-range ultrasonic signals. 

## Figures and Tables

**Figure 1 micromachines-14-00210-f001:**
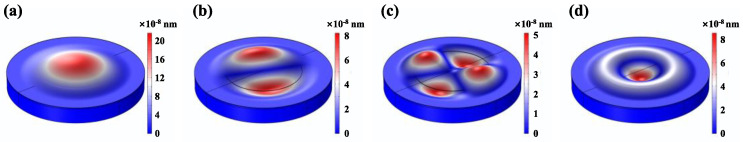
(**a**–**d**) The front four vibration modes of PMUT.

**Figure 2 micromachines-14-00210-f002:**
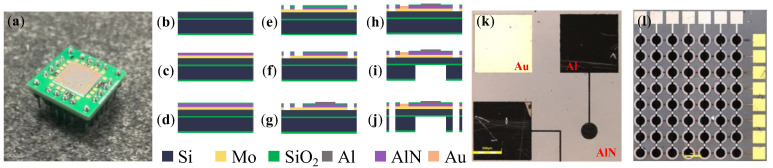
(**a**) PMUT device; (**b**–**j**) MEMS process flow; (**k**) PMUT cell; (**l**) PMUT array.

**Figure 3 micromachines-14-00210-f003:**
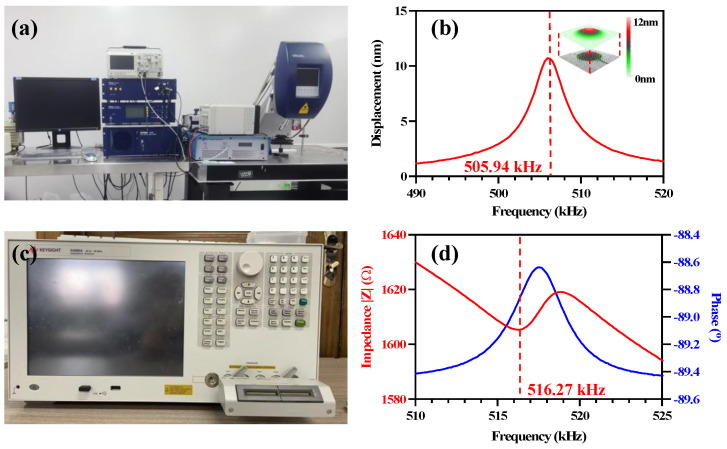
(**a**) The single-point Laser Doppler Vibrometer; (**b**) the displacement–frequency response; (**c**) the impedance analyzer; (**d**) the impedance testing of the PMUT.

**Figure 4 micromachines-14-00210-f004:**
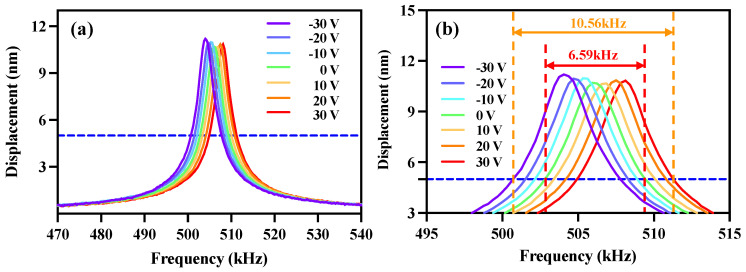
The frequency response curves of the PMUT array for a DC bias voltage range of −30 to 30 V: (**a**) full view; (**b**) local view.

**Figure 5 micromachines-14-00210-f005:**
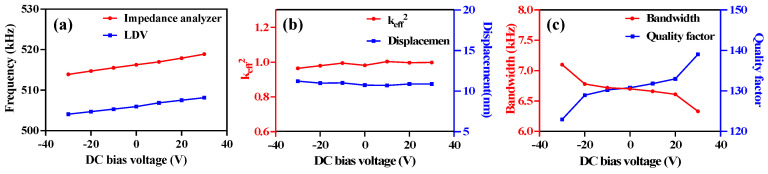
The DC bias effects on the performance of the PMUT: (**a**) the linear variation in frequency for estimating the effect of DC bias; (**b**) the effect of DC bias on $k_{eff}^{2}$ and film displacement; (**c**) the dependency of the −6 dB bandwidth and the Q on DC bias voltage.

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
