# Peer review of "Fabrication and DC-Bias Manipulation Frequency Characteristics of AlN-Based Piezoelectric Micromachined Ultrasonic Transducer"

_micromachines, 2023, doi:10.3390/mi14010210_

Round 1

Reviewer 1 Report

Comments to authors,

In the manuscript titled as “Fabrication and DC-bias manipulation frequency characteristics of AlN-based piezoelectric micromachined ultrasonic transducer”, the authors proposed a piezoelectric micromechanical ultrasonic transducers (PMUTs) based on AlN thin films that fabricated by standard MEMS process.

Meanwhile, the property characterizations, such as FEM simulation, impedance frequency spectrum, vibration displacement frequency spectrum, DC bias dependence of resonate frequency, effective coupling coefficient and Quality factor, etc., have been investigated. DC bias voltage is proven to be effective in the manipulation of AlN thin films based PMUTs, where the bandwidth is enlarged by 60.2%. This manuscript has some significance in terms of practical applications of piezoelectric micromechanical ultrasonic transducers (PMUTs). Yet, the following issues are necessary to be raised before its publication.

1.        In Fig 3(d), the vertical coordinate label should be “Impedance”, rather than “Impendance”.

2.        As for the Quality factor calculation, why is -6 dB bandwidth is used? As I know, -3 dB bandwidth is widely used in the calculation of Quality factor.

3.        In Page 6, “?eff2 is calculated to be 0.97% for this device”, what is the ?eff of other similar PMUTs? Is the ?eff2 of 0.97% superior to other PMUTs?

4.        In Page 6, “the quality factor of PMUT is intrinsically affected by air damping, anchor loss, thermoelastic dissipation and surface loss [33].” There is no reference 33, the author should check the reference numbers carefully.

5.        In Page 5, “a sensitivity of 68 Hz/V according to the LDV results”, “For the Impedance Analyzer results, a modification 183 of 4.98 kHz with a sensitivity of 83 Hz/V was obtained”. Why the sensitivities based on LDV results and Impedance Analyzer results are different? What is the actual sensitivity in device application?

6.        DC bias voltage can enlarge the bandwidth of PMUT, and the authors declare that such manipulation play an important role in underwater 3D imaging and nondestructive testing, the application scenarios need to be further specific.

In general, this manuscript still has some problems, and it needs minor revisions.

Reviewer 2 Report

Zhang et al. reported about the fabrication and DC-bias manipulation frequency characteristics of AlN-based piezoelectric micromachined ultrasonic transducer. I have some comments given below:

(1) Can authors describe why does the DC bias dependent performances of the device important in this study?

(2) Authors described that when DC bias voltage is applied to PMUT, the resonant frequency and bandwidth can be manipulated efficiently. Can authors explain why the application of DC bias facilitates in manipulating the frequency and bandwidth? What’s the corresponding mechanism in piezoelectric MEMS?

(3) In figure 1 in COMSOL simulation, there should be some color bar. Without that it does not make any sense. Importantly there is no detail descriptions of the figure 1.  

(4) What is the piezoelectric figure of merit (FoM) of the MEMs device?

(5) What is the pattering effect on the mechanical vibration of PMUT? Why do the authors choose this pattern during MEMS fabrication?

Reviewer 3 Report

In this article, the authors prepared Piezoelectric Micromachined Ultrasonic Transducer (PMUT) devices based on Aluminum Nitride (AlN) films which have enlarged BW and DC bias manipulated resonant frequency. The results show that the center frequency, electromechanical coupling coefficient, and -6 dB bandwidth of the devices were almost linear varying with a DC bias voltage from -30 V to 30 V. A modification of 4.08 kHz of resonant frequency and bandwidth enlargement of 60.2% was achieved due to the applied DC bias. These results show the potential of such devices used in ultrasonic imaging and nondestructive testing. The logic of this manuscript is clear. However, there are still some problems that need to be addressed before going to possible publication in this journal. 

1. Why did the authors choose applied DC bias voltage from -30 V to 30 V? Is it possible to expand the voltage range?

2. It would be better to give statistical data points in figure 5.

3. Line 144 ‘The surface profile of PMUT is shown in the insert of Figure 2(b)’ would be wrong.

4. The English language needs to check carefully in the revision stage because there are many careless mistakes in this manuscript.

5. The format of the reference needs to be double-checked.

Round 2

Reviewer 2 Report

The authors addressed the comments and made necessary changes in the manuscript. Now, it can be accepted in the present form.

Author Response

We would like to acknowledge the editor and reviewers for having spent time on handling and reviewing this manuscript and we appreciate the editor and referees’ comments that have helped us to improve the manuscript. We believe that the refereeing process has allowed us to improve the readability of our work. 
